# Shielding Capability Research on Composite Base Materials in Hybrid Neutron-Gamma Mixed Radiation Fields

**DOI:** 10.3390/ma16052084

**Published:** 2023-03-03

**Authors:** Ming Xiao, Qingao Qin, Xin He, Fei Li, Xiaobo Wang

**Affiliations:** 1College of Nuclear Technology and Automation Engineering, Chengdu University of Technology, Chengdu 610059, China; 2Applied Nuclear Technology in Geosciences Key Laboratory of Sichuan Province, Chengdu 610059, China

**Keywords:** radiation monitoring system, mixed neutron-gamma radiation, Monte Carlo method, shielding rate, radiation protection efficiency, mass attenuation coefficient

## Abstract

The ^16^N monitoring system operates in a mixed neutron-gamma radiation field and is subject to high background radiation, thus triggering instability in the ^16^N monitoring system measurement data. Due to its property of actual physical process simulation, the Monte Carlo method was adopted to establish the model of the ^16^N monitoring system and design a structure-functionally integrated shield to realize neutron-gamma mixed radiation shielding. First, the optimal shielding layer with a thickness of 4 cm was determined in this working environment, which had a significant shielding effect on the background radiation and improved the measurement of the characteristic energy spectrum and the shielding effect on neutrons was better than gamma shielding with the increase in the shield thickness. Then, functional fillers such as B, Gd, W, and Pb were added to the matrix to compare the shielding rates of three matrix materials of polyethylene, epoxy resin, and 6061 aluminum alloy at 1 MeV neutron and gamma energy. The shielding performance of epoxy resin as the matrix material was better than that of the aluminum alloy and polyethylene, and the shielding rate of boron-containing epoxy resin was 44.8%. The γ-ray mass attenuation coefficients of lead and tungsten in the three matrix materials were simulated to determine the best material for the gamma shielding performance. Finally, the optimal materials for neutron shielding and gamma shielding were combined, and the shielding performance of single-layer shielding and double-layer shielding in mixed radiation field was compared. The optimal shielding material-boron-containing epoxy resin was determined as the shielding layer of the ^16^N monitoring system to realize the integration of structure and function, which provides a theoretical basis for the selection of shielding materials in a special working environment.

## 1. Introduction

In order to ensure the safe operation of nuclear power plants, multiple real-time monitoring barriers have been set up. In the decades since the successful development of the ^16^N radiation monitoring system until it was put into use, most of the detectors were NaI (Tl+^241^Am), which provide guarantees for safe reactor operation due to lower maintenance costs and better energy resolution [1,2]. Additionally, the detectors need to be filled with adiabatic material and completely sealed with stainless steel to prevent short periods of high temperature and humidity that can occur during accident conditions [3]. The high ambient background radiation and operating temperature to which the detector is subjected affects the sensitivity of the monitoring device and leads to changes in the performance of certain components, which results in a drift of the measured γ-energy spectrum, leading to reference peak failures and other failures caused by them, accounting for 23% of the total failures [4,5]. To reduce the background radiation of neutron gamma rays in the operating environment of this system, the response speed was improved and the leakage was effectively determined. The peaceful reactor modification scheme was proposed by Yong Wang et al. [6] using MCNP modeling, where the scintillator was aligned to the monitored secondary cooling water pipeline and the detector assembly was housed in an aluminum enclosure and covered with a 5 mm lead shield, simulating the detector efficiency of the ^16^N monitoring system and effectively improved the reactor safety. However, in practical applications, a lead layer of only 5 mm does not block the background radiation of neutron gamma in harsh environments [7], therefore, for complex neutron gamma hybrid shielding, a series of composite materials with either excellent shielding performance or excellent mechanical properties, or high service life can be prepared by doping different functional fillers into the matrix material based on various types of matrices [8,9]. El-Sayed et al. [10] investigated the attenuation performance of titanite/epoxy composites against neutrons and γ-rays and found that composites with 80% titanite addition had more efficient thermal neutron and γ-ray attenuation performance at 500 mm thickness. Issa et al. [11] studied the multiple shielding properties of xPbO-(99-x)B_2_O_3_-Sm_2_O_3_ glass against gamma, neutrons, and protons, and concluded that the increase in the lead monoxide improved the radiation shielding performance of the glass samples. Akn [12] used Geant4 Monte Carlo simulation and Phy-X software to investigate the use of Ag_2_O–V_2_O_5_–MoO_3_–TeO_2_ as a radiation shielding material and the bioconcrete shielding usability of the material, respectively. In recent years, more and more studies have shown that composite shielding materials are more effective than traditional single shielding materials [13,14,15,16] and can be applied to a variety of special applications; in particular, boron oxide [17,18], gadolinium oxide [19], tungsten oxide [20], and lead oxide have been studied for neutron gamma shielding materials.

In this paper, Monte Carlo simulations were used to compare the shielding performance between different thicknesses and elements to facilitate the selection of shielding elements for a mixed irradiation environment. Among the commonly used neutron-gamma integrated shielding materials, polyethylene and epoxy resins have the advantages of good shielding, stable chemical properties, light weight, and small size as polymeric materials often used in the field of radiation protection; aluminum alloys are stronger and can easier meet the mechanical properties of the material, in addition to good corrosion resistance, heat resistance, and low density. The shielding rates of polyethylene, epoxy resin, and 6061 aluminum alloy matrix materials doped with boron, gadolinium, tungsten, and lead were compared to determine the appropriate thicknesses and optimize effective shielding body materials that can reach both neutron and gamma rays to achieve structural/functional integration, reduce the background interference in the complex radiation environment of the ^16^N radiation monitoring system, improve the response speed and effectively determine the amount of leakage.

## 2. Principle and Calculation Method

### 2.1. Leakage Rate Monitoring Principle of Steam Generator Tube

During normal operation of a nuclear power plant, the oxygen atoms in the primary coolant are irradiated by fast neutrons above 10 MeV, producing an intrinsic activation product ^16^N with energies of 7.12 MeV (5%) and 6.13 MeV (69%), respectively. Under normal operating conditions, it is only possible to measure the radionuclide ^16^N in the primary system. If the heat transfer tube is damaged, ^16^N will leak out into the secondary circuit with the coolant and mix with the pipeline steam. Therefore, the greater the γ-ray count rate generated by ^16^N decay in the main steam pipeline, the higher the leakage rate of the heat transfer tube of the steam generator. In practical applications, the ^16^N radioactivity level (γ count rate) in the main steam is frequently detected directly, and the damage of the heat transfer tube is determined by formula conversion [21,22,23,24].

### 2.2. Model Construction

The composition and thickness of the shielding material of the detector [25,26,27,28] will affect the shielding effect of the material on nuclear radiation. If the shielding effect of various factors on the material is studied through the experimental results, it will result in huge human and material consumption. Therefore, Monte Carlo simulation experiments can not only provide a theoretical basis for sample design and preparation, but also save cost and time [29,30]. The model of the NaI detector to detect the leakage of the steam generator heat transfer tube was constructed by the Monte Carlo method, which mainly includes the setting of the source term in the steam pipe, the structure of the detector, and the geometric position relationship between them.

As shown in Figure 1, the geometric position construction principle of the source term and the detector is to establish a three-dimensional coordinate axis with the center of the bottom of the detector. The center is 52 cm away from the center of the steam pipe.

The source term is the steam containing ^16^N radionuclides in the main steam pipe. The length of the main steam pipeline is 400 cm, the inner radius of the pipeline is 36.83 cm, the outer radius of the pipeline is 43.53 cm, and the thickness of the steam pipeline is 4.47 cm considering the weakening effect of the three-layer shielding of steam, the pipe wall, and insulation material outside the pipe wall on the source term.

The structure of the detector is shown in Figure 2. The main detector structure adopts a NaI detector, which is composed of the NaI crystal with a diameter of 3 inches and a thickness of 3 inches, optical glass, reflective layer, photomultiplier tube, and peripheral aluminum shell. Because the working temperature of the detector usually has a range of 30–55 °C, and the temperature may be high for a short time with high humidity under accident conditions, a large number of aluminum silicate insulation materials are filled in the detection device, and the stainless steel shell is completely sealed.

### 2.3. Radiation Protection Efficiency

In the application of nuclear technology, according to the situation of different radiation sources passing through the material, a special structure and specific shielding radiation shielding materials are used to protect the human body and equipment [31,32]. Neutron shielding slows neutrons into thermal neutrons by inelastic scattering and elastic scattering with matter, and is finally captured and absorbed by matter. Gamma rays lose energy through three effects: the photoelectric effect, Compton effect, and electron pair effect.

The shielding performance of the shielding material for neutron and gamma rays can be expressed by the radiation protection efficiency (*RPE*) [33]:(1)$$RPE=1-\frac{I}{I_{0}}\cdot 100\%$$
where *I* and *I_0_* are the dose after passing through the shielding material and the dose before passing through the shielding material, respectively, and the shielding material outgoing surface *I* and incoming surface *I_0_* are obtained by Monte Carlo simulation. The better shielding efficiency of the material indicates greater shielding of the incident particles and therefore better radiation protection. The larger the *RPE* value, the better the protective effect of the material, and vice versa.

### 2.4. Mass Attenuation Coefficient

The mass attenuation coefficient (μρ) of different shielding materials can be calculated by simulating the γ-ray transmission at different energies. When the γ-ray passes through a certain thickness medium, the attenuation mass coefficient is described by Equation (2) [34].
(2)I=I0⋅e−(μρ)ρ⋅x
where *I* and *I_0_* are the shielding material outgoing surface and incoming surface above-mentioned, respectively; μ is the linear attenuation coefficient of the medium; ρ and *x* are the linear attenuation coefficient and thickness of the shielding material, respectively. For a compound or mixture, its mass attenuation coefficient can be expressed as:(3)μρ=∑inωi(μρ)i=ω1(μρ)1+ω2(μρ)2+…+ωn(μρ)n
where ω_i_ and (μρ)i are the mass fraction and mass attenuation coefficient of the *i* th element or mineral, respectively.

## 3. Result and Discussion

### 3.1. Shield Thickness

As shown in Figure 1, the shielding layer with the special structure could not only help the detector shield the interference rays in the working environment, but also reduced the radiation damage of the detection system and prolonged the service life. Therefore, the thickness of the shield is very important in the selection of the neutron gamma composite field, the best shielding effect, the best material cost, and the response efficiency of the detector. In the ^16^N monitoring system, the distance between the detection device and the steam pipe was only about 4 cm. The model of the NaI detector to detect the leakage of the steam generator heat transfer tube was established, and the energy spectrum measurement without the shielding layer and with a 4 cm shielding layer was compared.

Figure 3 compares the energy spectrum measurement without the shielding layer and after adding a 4 cm tungsten shielding layer. It can be seen from the figure that the addition of the tungsten shield had a more obvious effect on the attenuation of low-energy gamma, showing a significant reduction in the height of the pulse obtained by simulation in the energy band less than 0.5 MeV; in the energy band 5–6.5 MeV, the decrease in the full-energy peak, single escape peak, and double escape peak of the two energy spectra was small, and the impact on the gamma shielding in the high energy band was small. After the spectrum was smoothed, the detection efficiency corresponding to the 6.13 MeV full energy peak was calculated to be 2.748 × 10^−2^ cps/Bq and 3.006 × 10^−2^ cps/Bq, respectively. It can be concluded that the tungsten shield with a thickness of 4.0 cm had an obvious shielding effect on the background radiation in the working environment. In other studies conducted in [35,36], similar results were obtained regarding the increase in the shield thickness. Selection of an appropriate shield thickness can reduce the contribution of low-energy gamma to the energy spectrum measurement and improve the characteristic energy spectrum measurement of the ^16^N radiation monitoring system, reducing the external interference, and improving the stability of the system.

Neutrons can be slowed down by inelastic collisions with heavy elements. After reducing to a certain energy, inelastic scattering will no longer occur, mainly through elastic scattering with hydrogen elements to lose energy [37,38,39]. Therefore, tungsten and lead, as commonly used heavy metal gamma shielding materials, can be used as a comprehensive shielding material for high-energy neutron slowing and gamma rays. In this paper, the background radiation below 1 MeV was shielded, and the shielding rates of lead and tungsten with a thickness of 0–4 cm to 1 MeV neutron and gamma energy were compared.

Figure 4 shows the simulation of the shielding rate of neutron and gamma rays when the thickness of the lead and tungsten shielding materials was 0.5 cm, 1 cm, 1.5 cm, 2 cm, 2.5 cm, 3 cm, 3.5 cm, and 4 cm, respectively. As can be seen in Figure 4a,b, the *RPE* values of lead and tungsten with different thicknesses for neutron and gamma rays were different. With the increase in the thickness, the radiation protection efficiency of the material started to increase at a faster rate and then slowed down, because the thicker material caused more collisions between the incident particles and the nucleus when passing through the shielding material, which decayed exponentially with the depth of incidence. However, as the shield thickness increased, the radiation protection efficiency against gamma tended to be more and more similar to that of neutron radiation protection, which verified Zhao Sheng’s [40] study on the thickness and shielding performance, where the improvement in the shielding performance against neutrons was obvious at thinner thicknesses, and the shielding performance against gamma and neutrons tended to be the same when the shield reached a certain thickness. The reason why the comprehensive shielding performance of tungsten against neutron gamma is much greater than that of lead shielding is because the mass density of tungsten is 19.35 g/cm^3^ greater than that of lead at 11.34 g/cm^3^, which has a larger cross-sectional action with incident particles and more collisions. The *RPE* values of both increase as the thickness of the shielding material increases (i.e., the better the shielding effect on the incident particles) [41,42]. At a thickness of 4 cm, both lead and tungsten achieved 30% shielding for neutron and gamma radiation shielding, so the simulated shielding thickness was determined to be 4 cm.

### 3.2. Shielding of Different Radiation Fields

Traditional neutron shielding materials such as boron, water, polyethylene, gadolinium [43,44,45,46,47] and gamma shielding materials such as lead, iron, tungsten, etc. [48,49,50] have a certain protective effect when dealing with a single radiation type. However, in the practical application of radiation protection scenarios, the complex and harsh space environment and radiation environment make the radiation shielding materials not only require excellent thermodynamic performance, but also take into account the shielding of neutron and gamma mixed radiation fields. Therefore, the common matrix materials polyethylene, epoxy resin, and 6061 aluminum alloy were selected. According to the three radiation environments of gamma radiation field, neutron radiation field, and neutron–gamma mixed radiation field, four different functional fillers such as B, Gd, W, and Pb were doped into the matrix material, which can not only reduce the weight of the shielding body, but also improve the shielding performance.

Figure 5 shows the radiation protection efficiency (*RPE*) of composite polyethylene materials with different proportions of B, Gd, W, and Pb functional fillers in the gamma radiation field, neutron radiation field, and neutron–gamma mixed radiation field. The horizontal coordinate is the content of the functional filler in the matrix and the vertical coordinate indicates the radiation protection efficiency (*RPE*) of the material. The *RPE* when the share ratio is 0 corresponds to the shielding rate of the pure matrix material. Where y = 0.25” in Figure 5 indicates that the material has a radiation protection efficiency of 25%. It can be seen from Figure 5a that in gamma shielding, with the increase in four functional fillers, the *RPE* of the composite polyethylene materials to gamma particles also showed different upward trends. As the content increased, the polyethylene composites doped with lead and tungsten had a greater effect on the gamma particle shielding performance than the polyethylene composites doped with gadolinium and boron. The *RPE* of W and Pb was always greater than that of B and Gd. Figure 5b,c represents the neutron radiation field and the mixed radiation field, respectively. Due to the high thermal neutron absorption cross section of elemental boron, some boron-containing compounds such as boron carbide (B_4_C) [51], boric acid (H_2_BO_3_) [52], boron nitride (BN) [53], or boron monomers have been widely used as thermal neutron absorbing functional fillers. Both showed that only for the boron-containing composites with an increase in the boron element, the radiation protection efficiency of their composites gradually increased and the shielding performance was significantly improved. Because the atomic number of Gd, W, and Pb is too large, the neutron energy of 1 MeV is not sufficient to reach a certain threshold and inelastic collision occurs, which cannot achieve good neutron shielding, only with the H element in polyethylene elastic collision with the increase in the filler element. Thus, the H element in the composite material is reduced, so the neutron cross section is reduced, and instead of enhancing the shielding performance of the polyethylene material against neutrons, the shielding against neutrons is reduced. The results show that the polyethylene matrix material is more suitable for use in neutron radiation protection, and that there is a higher neutron gamma shielding performance with an increase in the boron element content in the polyethylene matrix.

As shown in Figure 6, the radiation protection efficiency (*RPE*) of four fillers with 50% incorporation in polyethylene, epoxy resin, and 6061 aluminum alloy matrix materials were compared in the neutron radiation field. Among them, pure epoxy resin *RPE* > pure 6061 aluminum alloy *RPE* > pure polyethylene *RPE*; after adding four kinds of functional fillers, the shielding performance of the composite shielding material with 50% B element was better than that of the composite shielding material composed of the other three elements. The shielding rates of boron-containing polyethylene, boron-containing epoxy resin, and boron-containing aluminum alloy were 37.5%, 44.8%, and 40.5%, respectively. Therefore, the composite material composed of epoxy resin as the matrix in the neutron radiation field can be better shielded, and the incorporation of boron can better shield neutrons.

### 3.3. Mass Attenuation Coefficient of Different Shielding Materials

To determine the best gamma shielding material, the transmission calculation models of lead, tungsten, lead/polyethylene, tungsten/polyethylene, lead/epoxy resin, tungsten/epoxy resin, lead aluminum alloy, and tungsten aluminum alloy to γ-ray were established. The mass attenuation coefficients of different shielding materials were calculated under 0.2 MeV, 0.5 MeV, 1.0 MeV, 1.5 MeV, 2.0 MeV, 3.0 MeV, 4.0 MeV, 5.0 MeV, and 6.0 MeV gamma energy.

According to Formula (3) and the proportion and density of each material, the influence of Pb and W shielding particles on the mass attenuation coefficient of shielding materials in Figure 7 can be obtained by the simulation calculation. As can be seen from Figure 7, the mass attenuation coefficient of each shielding material tended to decrease as the photon energy increased, showing a better shielding performance for low-energy photons and a lower shielding effect for high-energy photons. When the photon energy was less than 1 MeV, the mass attenuation coefficient of the elemental lead and lead-containing composite materials was higher than that of elemental tungsten and tungsten-containing composite materials, and the composites containing tungsten had the lowest mass attenuation coefficient. This is because the atomic number of lead is greater than the atomic number of tungsten. In this energy range, more photoelectric effects and Compton scattering effects occur with photons to shield them. The atomic number in a material determines the shielding efficiency for low-energy photons, so the lead shielding effect is better than tungsten shielding. When the photon energy was greater than 1 MeV and less than 2 MeV, three effects combined to influence the shielding of photons, and the photon shielding effect of lead-containing materials and tungsten-containing materials was not much different. When the photon energy was greater than 2 MeV, the action cross section was reduced and only an electron pair effect could take place with the target atom, subsequently producing two 0.511 MeV photons that are then shielded by the photoelectric and Compton effects. The difference in shielding performance between the various types of matrix materials containing lead and tungsten was apparent: polyethylene matrix > epoxy resin matrix > aluminum alloy matrix. The mass attenuation coefficient of the tungsten-containing materials was greater than that of the lead-containing materials, that is, the shielding effect of the tungsten-containing materials was better than that of the lead-containing materials, because in the study of actual radiation shielding protection, the shielding effect of materials mainly depends on the absorption cross section of the shielding composition, material thickness, and density. When the shielding thickness was the same, the density of tungsten was greater than that of lead. The reaction cross section of the electron pair effect between the photon and tungsten was larger, so the shielding effect was better. Therefore, for the research environment of this paper, for gamma photon energies less than 1 MeV, lead-containing materials can be selected for effective shielding against gamma radiation.

### 3.4. The Difference between Single-Layer Shielding and Double-Layer Shielding Materials in a Mixed Radiation Field

By calculating the neutron and gamma ray shielding properties of the three kinds of matrices and four kinds of functional shielding fillers, the material designed with epoxy resin as the matrix and 50% B element as the shielding filler can be selected as the neutron shielding composite. When the gamma photon energy was less than 1 MeV, the mass attenuation coefficient of the lead and lead-containing composites was higher than that of the tungsten and tungsten-containing composites, and the relationship between the single-layer shielding composites and double-layer shielding composites and shielding properties was further studied.

The variation in the shielding rate of single-layer shielding and double-layer shielding materials with the material composition is shown in Table 1. The double-layer shielding combination can be divided into two categories: A and B according to the thickness of the material before and after, which were 2 cm + 2 cm and 3 cm + 1 cm, respectively. In the two combinations of A and B, the shielding rate of the combination of A was lower than that of the combination of B. After adding the shielding layer, the shielding rate was less than 0, and the number of shielded particles did not decrease but increase. This is because at the thickness of 2 cm + 2 cm, the probability of interaction between neutrons and effective elements such as boron and hydrogen elements decreased, and the remaining neutrons and gamma rays and secondary gamma rays generated by neutron protection could not be effectively shielded. In the class B combination, the thickness of 3 cm could increase the probability of collision between neutrons and effective elements, the latter 1 cm of material providing shielding against the secondary gamma or residual neutrons generated in front of it. Therefore, the shielding performance of the class B combination material was higher than that of the class A combination. In the two double-layer shielding combinations of A and B, incident neutrons collided with the element boron several times to reduce energy, after which secondary gamma radiation shielding was achieved through the lead layer, resulting in a comprehensive gamma neutron shielding that makes the boron-containing epoxy resin plus lead combination have a higher shielding rate than other material combinations. The difference in the thickness of the boron-containing epoxy resin in the two double-layer shielding combinations A and B resulted in the two showing significant differences in radiation protection efficiency. Among the single-layer shielding material combinations, the single lead, tungsten, boron, and epoxy resin and the shielding materials of these four raw material combinations are listed. It can be seen that the neutron gamma shielding performance of monolithic tungsten and monolithic boron was excellent, followed by the combined shielding materials, and compared with the double-layer shielding composite material, the shielding rate of the single-layer shielding composite material with a thickness of 4 cm was much larger than that of the double-layer shielding composite material, that is, the shielding performance was better than the double-layer shielding combination. Although both monolithic tungsten and monolithic boron have excellent conditions for comprehensive gamma neutron shielding, tungsten is too dense and expensive to be widely used, and there is no free boron in nature, and artificially prepared boron monomers are not stable, so in practical applications, in order to make the material lightweight and have excellent shielding performance, people more often use composite materials. In Table 1, the radiation protection efficiency of the epoxy resin was 39.114%, with a good neutron gamma shielding effect and a melting point of 115–120 °C. By using it as a matrix and mixing it with functional fillers, the material can be lightweight and have excellent shielding performance. Therefore, at a neutron energy of 1 MeV, the highest radiation protection efficiency of 44.8% was derived for the boron epoxy resin composite containing 50% boron.

## 4. Conclusions

In this paper, the working environment of the detector was affected by neutron–gamma mixed radiation, which affects the sensitivity of the monitoring equipment. A shielding material with an appropriate thickness was designed to ensure the detection efficiency and stability of the detection system by reducing the background interference rays in the environment to reduce the failure probability of the ^16^N monitoring system.

The model of the ^16^N monitoring system was established by the Monte Carlo method, by comparing the energy spectra measured without and with the addition of a 4 cm tungsten shield, and spectral smoothing of the energy spectra revealed the detection efficiencies of 2.748 × 10^−2^ cps/Bq and 3.006 × 10^−2^ cps/Bq for the 6.13 MeV full energy peak, respectively. Therefore an appropriate shield thickness was chosen to reduce the contribution of secondary gamma to the energy spectrum measurements, and the optimal shielding layer of 4 cm thickness was determined, which had an obvious shielding effect on the background radiation and improved the measurement of the characteristic energy spectrum. Comparing the shielding rate of lead and tungsten with the thickness of a 0–4 cm shielding layer, when the thickness was 4 cm, the shielding rate of the neutron and gamma rays could reach about 30%.

Comparing the polyethylene matrix with different contents of boron, gadolinium, tungsten, and lead, the results demonstrate that in the gamma environment, with the increase in the filler content, the polyethylene composites doped with lead and tungsten had a greater effect on the shielding performance of gamma particles than the polyethylene composites doped with gadolinium and boron; in the neutron environment, the increase in the functional fillers does not imply an enhancement in their shielding performance, and the increase in gadolinium, tungsten, and lead instead weakened the neutron gamma shielding effect of the composites; only the increase in the boron element enhanced the shielding effect of the polyethylene containing boron.

Comparing the three matrix materials of polyethylene, epoxy resin, and the 6061 aluminum alloy, the neutron shielding performance of epoxy resin was better than that of the aluminum alloy and polyethylene. Comparing the mass attenuation coefficients single lead and tungsten in the three matrix materials against γ-rays, it was determined that the shielding performance of the material containing lead was optimal at a gamma energy of 1 MeV. The relationship between the single-layer shielding composites and double-layer shielding composites and the shielding performance was investigated. The results demonstrate that at a neutron energy of 1 MeV, the shielding rate of single-layer shielding composites was much greater than that of the double-layer shielding composites, and it can be concluded that the radiation protection efficiency of the epoxy resin composite doped with boron and lead elements was 44.8%, and its shielding performance was better than that of the double-layer shielding materials of which it was composed. Therefore, the optimal shielding material-boron-containing epoxy resin can be used as the shielding layer of ^16^N monitoring system under the 1 MeV neutron gamma mixed radiation field.

## Figures and Tables

**Figure 1 materials-16-02084-f001:**
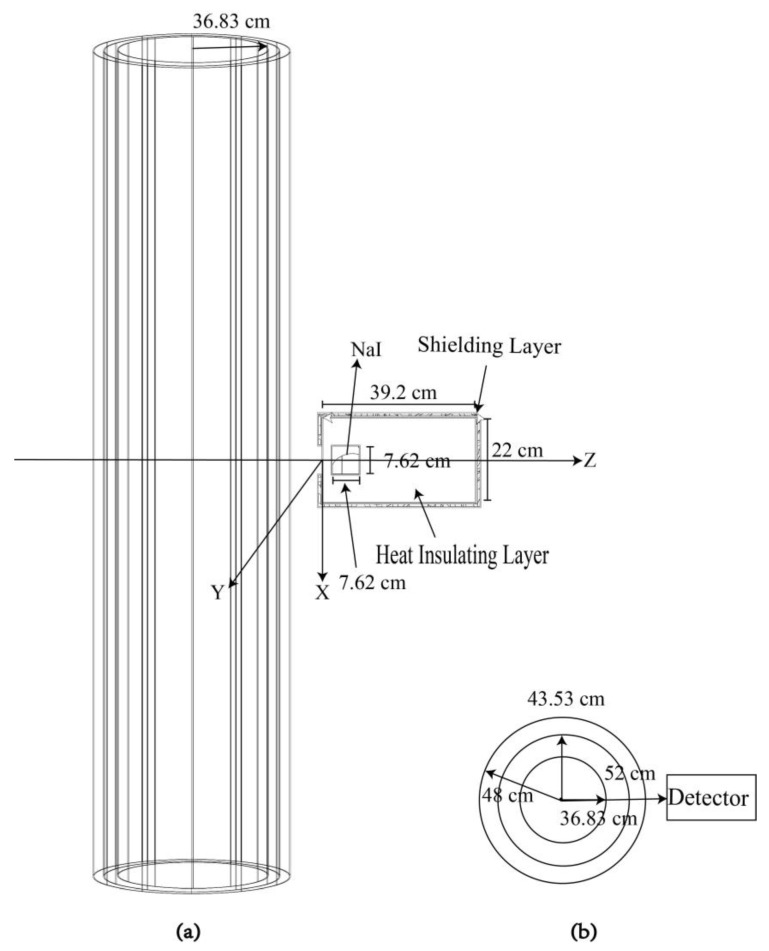
Position design of the detector and secondary circuit steam pipe. (**a**) Front view. (**b**) Side view.

**Figure 2 materials-16-02084-f002:**
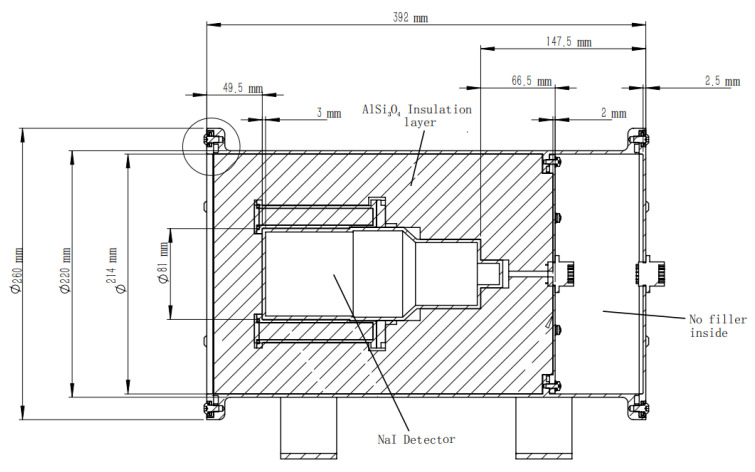
Structure diagram of the NaI detector.

**Figure 3 materials-16-02084-f003:**
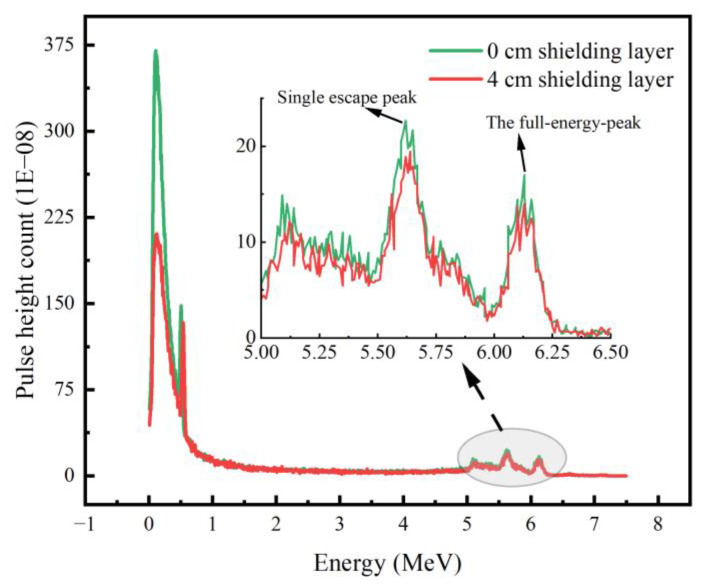
Energy spectrum measurement of the ^16^N radiation monitoring system without the shielding layer and with a 4 cm tungsten shielding layer.

**Figure 4 materials-16-02084-f004:**
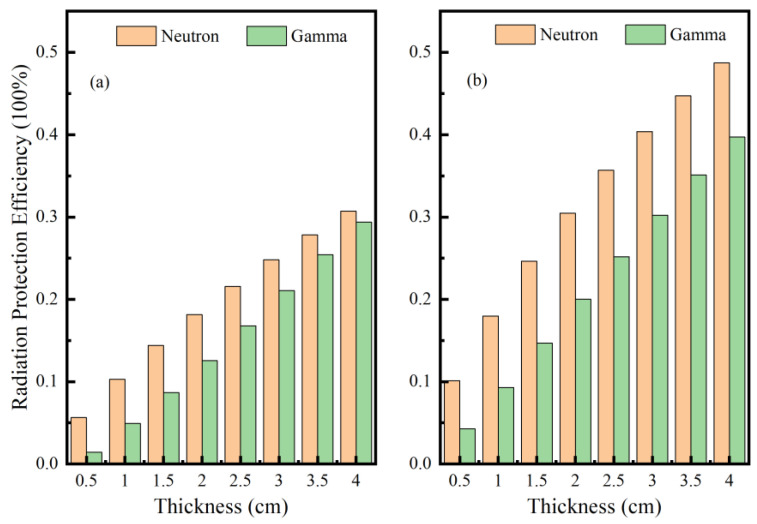
Radiation protection efficiency (*RPE*) of lead and tungsten to neutron and gamma rays at different thicknesses (**a**) Shielding rate of lead with increasing thickness; (**b**) Shielding rate of tungsten with increasing thickness.

**Figure 5 materials-16-02084-f005:**
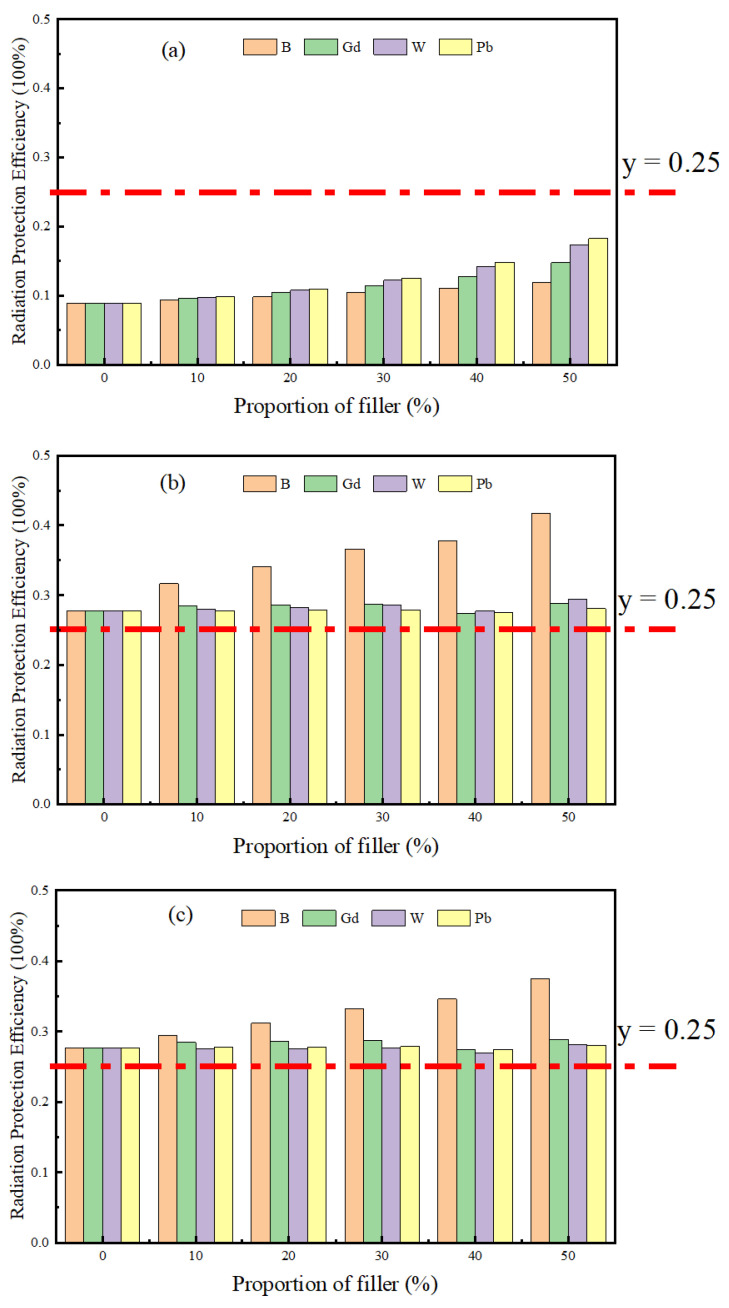
The radiation protection efficiency (*RPE*) of different proportions of functional filler/polyethylene in three radiation fields: (**a**) gamma radiation field; (**b**) neutron radiation field; (**c**) neutron gamma mixed radiation field.

**Figure 6 materials-16-02084-f006:**
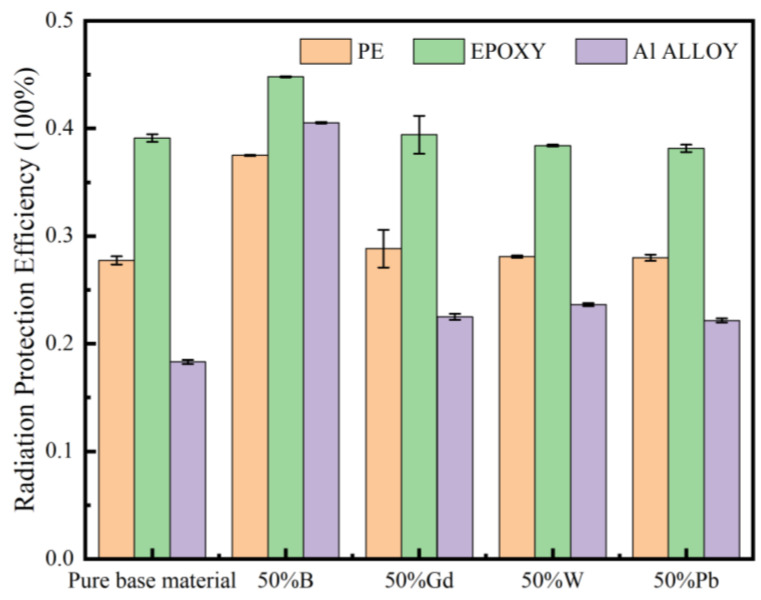
Radiation protection efficiency (*RPE*) of three matrix shielding materials in the neutron radiation field.

**Figure 7 materials-16-02084-f007:**
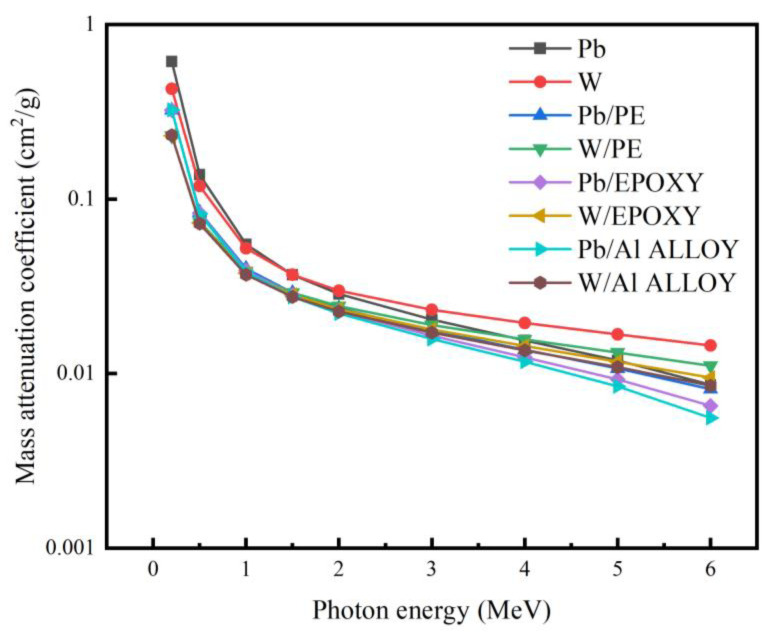
Mass attenuation coefficients of Pb, Wu functional fillers at 0.2 MeV–6 MeV gamma photon energies.

**Table 1 materials-16-02084-t001:** Radiation protection efficiency (*RPE*) of single-layer and double-layer shielding materials.

Double Layer Shielding	Structure Combination	A: 2 cm + 2 cm	B: 3 cm + 1 cm
	B/EP+W	−8.514%	11.973%
	W+B/EP	−2.895%	17.534%
	B/EP+Pb	1.796%	20.245%
	Pb+B/EP	−4.686%	10.355%
	B/EP+W/EP	−6.309%	15.466%
	W/EP+B/EP	−14.312%	−1.1347%
	B/EP+Pb/EP	−4.839%	16.523%
	Pb/EP+B/EP	−14.997%	0.707%
Single Layer Shielding	Material components		
	50%B + 50%Epoxy	44.8%	
	25%B + 25%Pb + 50%Epoxy	43.570%	
	25%B + 25%W + 50%Epoxy	43.866%	
	100%B	52.122%	
	100%W	48.716%	
	100%Pb	30.692%	
	100%Epoxy	39.114%	

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
