# Peer review of "Shielding Capability Research on Composite Base Materials in Hybrid Neutron-Gamma Mixed Radiation Fields"

_materials, 2023, doi:10.3390/ma16052084_

Round 1
Reviewer 1 Report
This work addresses shielding properties research on composite base materials in hybrid neutron-gamma mixed radiation fields. The authors mainly focused on the 16N monitoring system. They tried to decrease the background radiation to let the accurate data read by using different radiation shielding materials containing distinct dopants. The authors implemented a simulation work using the Monte Carlo method to perform the investigation. In addition to this, various thicknesses were used to estimate the radiation protection efficiency for the intended samples. Their findings showed that the optimal shielding material was obtained as boron-containing epoxy resin. In my opinion, the research field is highly popular nowadays; therefore, the author’s intention to develop a high-performance hybrid neutron-gamma protection material is considered valuable. Moreover, the combination material systems rely upon a scientific basement, and the authors are aware of the existing literature studies. With all these mentioned, I recommend its publication in your esteemed journal after the authors revised my suggestions listed below. My final decision is a major revision for this paper.
a. The abstract part should be composed of the main characteristics of the paper rather than explaining the research field. That is to say, the authors should shorten the first couple of sentences because the readers will comprehend the knowledge in the introduction part.
b. The authors should mention the dopants such as B, Gd, W, and Pb in the abstract part so that the readers will understand the scope of the present investigation.
c. The intended energy level should be mentioned in the abstract part while explaining the Monte Carlo method.
d. The authors shall add “Radiation protection efficiency” to the keywords.
e. In the introduction, the authors discussed some essential literature surveys to evaluate their intended material system. I think the authors should give some studies consisting of B2O3, Gd2O3, WO3, and PbO dopants’ effects on photon shielding and neutron absorption. The below-given literature studies shall be discussed therein.
• Kurtulus, R., Kavas, T., Calculations on Linear Attenuation Coefficient and Fast Neutron Removal Cross-section for B2O3-TeO2 Glass System via Phy-X/PSD, Journal of the Turkish Ceramic Society, 1 (4), 19-23, 2022
• Kurtulus, R., Kavas, T., Akkurt, I., Gunoglu, K.,Theoretical and Experimental Gamma-rays Attenuation Characteristics of Waste Soda-lime Glass Doped with La2O3 and Gd2O3, Ceramics International, vol. 47, 6, pp. 8432-8440, 2021
• Kurtulus, R., Kavas, T., The role of B2O3 in lithium-zinc-calcium-silicate glass for improving the radiation shielding competencies: A theoreticalevaluation via Phy-X/PSD, Journal of Boron, 6(1), 234-242, 2021
• Sayyed, M.I.; Hashim, S.; Hannachi, E.; Slimani, Y.; Elsafi, M. Effect of WO3 Nanoparticles on the Radiative Attenuation Properties of SrTiO3 Perovskite Ceramic. Crystals 2022, 12, 1602
f. The explanations in relation to the figures should be done before, not after the figures.
g. The equations should be rechecked because the variables were not thoroughly stated in the explanations, i.e., T, RPE, or the like, are missing. Besides that, the sub-indices should be checked throughout the manuscript file. Otherwise, it canalized me that the manuscript file was not concentratedly prepared.
h. The variables in the equations and in the text should be fonted similarly to the whole text. Some are bigger; some are smaller…
i. The zoomed area in Figure 3 should be arranged so that the difference can easily be read after adjusting the y-axis interval. It should be compassed between 0 and 25, not -10 and 40.
j. The obtained successful result in Figure 3 should be confirmed with the existing similar findings so that the authors can attribute their findings in an easier way.
k. In my opinion, there is no need to add lines in Figures 4 and 5. It is simple to track the incremental trend with the bar charts. Please remove them.
l. The parameter- RPE findings must be compared with the available literature results. The studies containing similar thickness values should be chosen, and all findings should be evaluated so that the results can be comprehended in a better way. Some literature findings can be found below:
• M. I. Sayyed, M. Kh. Hamad, M.H.A. Mhareb, Recep Kurtulus, Nidal Dwaikat, Marwa Aldikhel, Mohamed.Elsafi, Malaa M. Taki, Taner Kavas, Kh. A. Ziq, Mayeen Uddin Khandaker, D. A. Bradley, Assessment of Radiation attenuation properties for novel alloys: An experimental approach, Radiation Physics and Chemistry, Vol.200, 110152, 2022
• Kavas, T., Sultan J. Alsufyani, Z.A. Alrowaili, Abdulaziz N. Alharbi, Kurtulus, R., Oyeleke Olarinoye, M.S. Al-Buriahi, Influence of iron (III) oxide on the optical, mechanical, physical, and radiation shielding properties of sodium-barium-vanadate glass system, Optik, 257, 168844, 2022The y-axis intervals in Figure 5 should be re-considered, this is because the differences may not be properly observed. In the case of adjusting the ranges, one can see the bar charts better.
m. In general, I see a lack of discussions with the existing literature. Yet again, Figure 5 should be discussed in this sense. The authors only presented their findings and evaluated them with each other. However, it is critical to say that we should follow the existing literature and target to advance in that way.
n. To my way of thinking, Figure 7 data should be combined in one figure. The readers can understand the variations by observing all samples together, which in turn, it will pave the way for better comprehension.
o. More numerical findings should be given in the conclusion part. There are many valuable findings in the paper, so the authors should summarize these in the best way.
Author Response
Response to Reviewer 1:
Thanks for your precious time and constructive comments on our paper. The paper is revised according to your comments. The following is the answers and revisions we have made in response to the questions and suggestions on an item by item basis.
- The abstract part should be composed of the main characteristics of the paper rather than explaining the research field. That is to say, the authors should shorten the first couple of sentences because the readers will comprehend the knowledge in the introduction part.
We sincerely appreciate the valuable comments and we have simplified the first few sentences of the abstract. Rewritten new “ The 16N monitoring system operates in a mixed neutron-gamma radiation field and is subject to high background radiation, thus triggering instability in the 16N monitoring system measurement data ”, which can be seen in line 9-11.
- The authors should mention the dopants such as B, Gd, W, and Pb in the abstract part so that the readers will understand the scope of the present investigation.
We think this is an excellent suggestion. As suggested by the reviewer, we have added the functional fillers involved in this study, such as B, Gd, W and Pb, in line 17-18.
- The intended energy level should be mentioned in the abstract part while explaining the Monte Carlo method.
Thank you for the suggestion, we have added the energy levels of the Monte Carlo simulations used in this study in the abstract part, in line 19.
- The authors shall add “Radiation protection efficiency” to the keywords.
We sincerely thank you for careful reading. Radiation protection efficiency is indeed an important parameter for assessing the shielding performance of the material in this study, and we have added “Radiation protection efficiency”to the keywords, in line 30.
- In the introduction, the authors discussed some essential literature surveys to evaluate their intended material system. I think the authors should give some studies consisting of B2O3, Gd2O3, WO3, and PbO dopants’ effects on photon shielding and neutron absorption.
We realized the inadequacy of the literature review. According to your nice suggestion, we have reviewed the relevant literature you recommended and found it to be well suited to the content and direction of this paper. We have added references to studies on the effects of B2O3, Gd2O3, WO3 and PbO dopants on photon shielding and neutron absorption into the introduction part in the revised manuscript and cited References 17-20 to support this view.
[17] Kurtulus, R., Kavas, T., Calculations on Linear Attenuation Coefficient and Fast Neutron Removal Cross-section for B2O3-TeO2 Glass System via Phy-X/PSD[J].Journal of the Turkish Ceramic Society,2022,1 (4):19-23.
[18] Kurtulus, R., Kavas, T., The role of B2O3 in lithium-zinc-calcium-silicate glass for improving the radiation shielding competencies: A theoreticalevaluation via Phy-X/PSD[J].Journal of Boron,2021,6(1):234-242.
[19] Kurtulus, R., Kavas, T., Akkurt, I., et al. Theoretical and Experimental Gamma-rays Attenuation Characteristics of Waste Soda-lime Glass Doped with La2O3 and Gd2O3[J].Ceramics International,2021,47(6):8432-8440.
[20] M. I. Sayyed, S. Hashim, E. Hannachi, et al. Effect of WO3 Nanoparticles on the Radiative Attenuation Properties of SrTiO3 Perovskite Ceramic[J].Crystals,2022,12(1602): 1602.
- The explanations in relation to the figures should be done before, not after the figures.
Thanks for your reminder, We realized our mistake and swapped the position of Figures 1 and 2, shifting the previous Figure 2 explanation to better explain the design of the location of the detector and secondary circuit steam pipes.
- The equations should be rechecked because the variables were not thoroughly stated in the explanations, i.e., T, RPE, or the like, are missing. Besides that, the sub-indices should be checked throughout the manuscript file. Otherwise, it canalized me that the manuscript file was not concentratedly prepared.
We realized the inadequacy of this part. We have modified Equations 1 and 2. Considering that there is no discussion of transmittance (T) in the content of the results and discussion, Equation 1 has been removed and the variables that appear in the Radiation Protection Efficiency (RPE) equation have been explained in more detail, “Where I and I0 are the dose after passing through the shielding material and the dose before passing through the shielding material, respectively, and the shielding material outgoing surface I and incoming surface I0 are obtained by Monte Carlo simulation. The better the shielding efficiency of the material indicates more shielding of incident particles and therefore better radiation protection. The larger the RPE value, the better the protective effect of the material, and vice versa”, in line 133-142. It also explains the I and I0 in the mass attenuation coefficient equation, in line 144-145.
- The variables in the equations and in the text should be fonted similarly to the whole text. Some are bigger; some are smaller…
We feel sorry for our carelessness. In our resubmitted manuscript, we have revised the font of the variables in the equations and in the text to be similar to the font of the entire text. Thanks for your correction.
- The zoomed area in Figure 3 should be arranged so that the difference can easily be read after adjusting the y-axis interval. It should be compassed between 0 and 25, not -10 and 40.
We sincerely thank the reviewer for careful reading. As suggested by the reviewer, we have adjusted the y-axis spacing of the enlarged area in Figure 3 to make it easier for the readers to see the difference.
- The obtained successful result in Figure 3 should be confirmed with the existing similar findings so that the authors can attribute their findings in an easier way.
We feel great thanks for your professional review work on our article and we have added to Figure 3 in part, "Selection of an appropriate shield thickness can reduce the contribution of low-energy gamma to the energy spectrum measurement and improve the characteristic energy spectrum measurement of the 16N radiation monitoring system", and cite References 35 and 36 to support this view.
[35] Wang L, Lei M, Zhang W L et al. MCNP multi-modal calculation of detection efficiency of 16N γ-energy spectral power monitor[J]. Nuclear Electronics & Detection Technology,2020,40(01):109-114.
[36] Zhao S, Huo Z p, Zhong G q, et al. Preparation of modified gadolinium / boron / polyethylene nanocomposites and their shielding properties for neutron and gamma rays[J].Chemical Journal of Chinese Universities,2022,43(06):57-67.
- In my opinion, there is no need to add lines in Figures 4 and 5. It is simple to track the incremental trend with the bar charts. Please remove them.
We think this is an excellent suggestion. As suggested by the reviewer, we have censored the connecting lines in Figures 4 and 5 so that the differences between the bars can be more clearly seen.
- The parameter- RPE findings must be compared with the available literature results. The studies containing similar thickness values should be chosen, and all findings should be evaluated so that the results can be comprehended in a better way.
Thank for your suggestion. We are aware of the lack of comparison of the conclusions obtained with the existing literature. In the discussion of Figure 4, literature 41 and 42 have been added to support the view in order to better understand the study of the relationship between the parameter RPE and thickness. Also, literature 40 has been added to support the conclusion that "where the improvement in shielding performance against neutrons is obvious at thinner thicknesses, and the shielding performance against gamma and neutrons tends to be the same when the shield reaches a certain thickness".
[40] Zhao S. Preparation of modified gadolinium-boron polyethylene composites and study of neutron and gamma radiation shielding mechanisms[D]. University of Science and Technology of China, 2021. DOI:10.27517/d.cnki.gzkju.2021.002128.
[41] M. I. Sayyed, M. Kh. Hamad, M.H.A. Mhareb, et al. Assessment of Radiation attenuation properties for novel alloys: An experimental approach[J].Radiation Physics and Chemistry,2022,200:110152.
[42] Kavas, T., Sultan J., Z.A. Alrowaili, et al. Influence of iron (III) oxide on the optical, mechanical, physical, and radiation shielding properties of sodium-barium-vanadate glass system, Optik, 257, 168844, 2022
- The y-axis intervals in Figure 5 should be re-considered, this is because the differences may not be properly observed. In the case of adjusting the ranges, one can see the bar charts better.
We sincerely thank you for your careful reading. Due to our carelessness, we did not represent the y-axis ranges of the three smaller plots in Figure 5 consistently, which prevented the data from being displayed better. In response we have corrected the y-axis range in Figure 5(c). And to visualise the difference, a dashed line of "y=0.25" has been added to Figure 5 to indicate that the material has a radiation protection efficiency of 25 %.
- In general, I see a lack of discussions with the existing literature. Yet again, Figure 5 should be discussed in this sense. The authors only presented their findings and evaluated them with each other. However, it is critical to say that we should follow the existing literature and target to advance in that way.
Thank you very much for your valuable feedback, which made us aware of the lack of discussion with the existing literature. In the discussion in Figure 5, we have added information about the high thermal neutron absorption cross section of elemental boron to explain why only boron-containing composites have a progressive increase in radiation protection efficiency of their composites with increasing elemental boron. Literature 51, 52 and 53 are also introduced to support this view.
[51] Jiang D W, MURUGADOSS V, WANG Y, et al. Electromagnetic Interference Shielding Polymers and Nanocomposites-A Review[J]. Polymer reviews,2019,59(2):280-337. DOI:10.1080/15583724.2018.1546737.
[52] Kumar, P., Narayan M, Sikdar, A., et al. Recent Advances in Polymer and Polymer Composites for Electromagnetic Interference Shielding: Review and Future Prospects[J].Polymer Reviews,2019,59(4): 687-738
[53] Derradji M , Zegaoui A , Medjahed A , et al. Hybrid phthalonitrile‐based materials with advanced mechanical and nuclear shielding performances[J]. Polymer Composites, 2020.
- To my way of thinking, Figure 7 data should be combined in one figure. The readers can understand the variations by observing all samples together, which in turn, it will pave the way for better comprehension.
Thank you to the reviewers for their careful reading. As suggested, we have combined the data in Figure 7 into one figure. We have also added the mass attenuation coefficient as a function of photon energy to the discussion of Figure 7, detailing the reasons for the decrease in the mass attenuation coefficient of the material in each energy band. This enables the reader to fully understand the significance of Figure 7.
- More numerical findings should be given in the conclusion part. There are many valuable findings in the paper, so the authors should summarize these in the best way.
The authors realized the inadequacy of this part. According to your suggestion, we added the detection efficiencies corresponding to the full energy peaks without and with the addition of a 4 cm tungsten shield in the conclusion section, suggesting that selecting an appropriate shield thickness can minimise the contribution of secondary gamma to the energy spectrum measurements. Also, the differences in the shielding performance of the four fillers on the material in the gamma and neutron environments are added. Finally, the relationship between single and double shielding composites and shielding performance is summarised.

Reviewer 2 Report
The manuscript is fine and however it needs revision owing to following issues.
1. What is role of shield thickness on the shielding efficiency?
2. How do the radiation protection is connected to shielding efficiency.
3. For which sample of current work, the attenuation coefficient is low? Why?
4. Explain the dependence of attenuation coefficient on photon energy.
5. Are the polymers suitable for the shields in this work? Explain.
6. Table.1 needs much explanation as it plays vital role?
Author Response
Response to Reviewer 2:
Thanks for your precious time and constructive comments on our paper. The paper is revised according to your comments. The following is the answers and revisions we have made in response to the questions and suggestions on an item by item basis.
- What is role of shield thickness on the shielding efficiency?
Thank you for your question. We have made a supplementary discussion on Figure 4. As can be seen from the graph, the radiation protection efficiency (RPE) values for neutrons and gamma rays differ for different thicknesses of lead and tungsten. This is because the thicker material allows more collisions between the incident particles and the nucleus as they pass through the shielding material, which decays exponentially with the depth of incidence.
- How do the radiation protection is connected to shielding efficiency.
The better the shielding efficiency of the material indicates more shielding of incident particles and therefore better radiation protection. Relevant mentioned in line 136-137.
- For which sample of current work, the attenuation coefficient is low? Why?
Thanks to valuable comments, we have added the reasons for the low attenuation coefficient of the samples in the discussion in Figure 7. At low energy bands, the lowest mass attenuation coefficients are found for composites containing tungsten. This is due to the fact that at low photon energies, only the photoelectric and Compton effects occur with the material, so the atomic number in the material determines the efficiency of shielding against low energy photons. At intermediate energies, the composites exhibit the same mass attenuation coefficient and the three effects combine to affect shielding against photons. At high energies, the lead-containing composites have the lowest mass attenuation coefficients, and due to the high photon energy, the cross section of action is reduced and only an electron pair effect occurs with the target atom, subsequently producing two 0.511 MeV photons which are then shielded by the photoelectric and Compton effects.
- Explain the dependence of attenuation coefficient on photon energy.
The relationship between the attenuation coefficient and photon energy has been modified and supplemented in the discussion in Figure 7. As can be seen from Figure 7, the mass attenuation coefficient of each shielding material tends to decrease as the photon energy increases, showing better shielding performance for low energy photons and lower shielding for high energy photons.
- Are the polymers suitable for the shields in this work? Explain.
Thank you for your question. In this paper, we have used two polymers, polyethylene and epoxy resin, respectively. Both have the advantages of good shielding, stable chemical properties, light weight and small size as polymeric materials often used in the field of radiation protection and are supplemented in lines 73-75. Within the working environment covered in this paper, shielding materials need to be adapted to 30-55 ℃, while polyethylene and epoxy resins have melting points of 85-110 ℃ and 115-120 ℃ respectively. Therefore, we believe that it can be well adapted to this working environment as a shielding material.
- 1 needs much explanation as it plays vital role?
The authors realized the inadequacy of this part. According to your suggestion, we have added a discussion of the differences in thickness of the boron-containing epoxy resin in the two double shielding combinations A and B, resulting in the two showing significant differences in radiation protection efficiency. Also, to better explain the choice of composite shielding materials over monolithic shielding materials, we have added the radiation protection efficiency of monolithic lead, tungsten, boron and epoxy to the table. Although both monolithic tungsten and monolithic boron have excellent conditions for combined gamma neutron shielding, the high mass density and high cost of tungsten preclude their widespread use, and since free boron does not exist in nature and artificially prepared boron monomers do not exist stably, it is not practical to speak of monolithic tungsten and boron as shielding materials. Finally, it is concluded that the radiation protection efficiency of the epoxy resin composite containing 50% boron is the highest at a neutron energy of 1 MeV.

Round 2
Reviewer 1 Report
I thoroughly reviewed the revised manuscript. The authors seriously considered my suggestions, so they successfully revised the paper. In my opinion, this work will be of great interest by the scientific community . Therefore, this paper can be published in its present form.